# Inline Optical Coherence Tomography for Multidirectional Process Monitoring in a Coaxial LMD-w Process

**Charlotte Stehmar** [1,*], **Marius Gipperich** [1], **Markus Kogel-Hollacher** [2], **Alfredo Velazquez Iturbide** [1] and **Robert H. Schmitt** [1,3]

1 Fraunhofer Institute for Production Technology IPT, Steinbachstraße 17, 52074 Aachen, Germany; marius.gipperich@ipt.fraunhofer.de (M.G.); alfredo.velazquez.iturbide@ipt.fraunhofer.de (A.V.I.); r.schmitt@wzl.rwth-aachen.de (R.H.S.)

2 Precitec Optronik GmbH, Schleussnerstraße 54, 623263 Neu-Isenburg, Germany; m.kogel-hollacher@precitec.de

3 Laboratory for Machine Tools and Production Engineering (WZL), RWTH Aachen University, Campus-Boulevard 30, 52074 Aachen, Germany

* Correspondence: charlotte.stehmar@ipt.fraunhofer.de; Tel.: +49-241-8904-781

**Abstract:** Within additive manufacturing, process stability is still an unsolved challenge. Process instabilities result from the complexity of laser deposition processes and the dependence of the quality of the workpiece on a variety of factors in the process. Because a stable process is dependent on many different factors, permanent precise inline monitoring is required. The suitability of the optical coherence tomography (OCT) measuring system integrated into a wire-based laser metal deposition (LMD-w) process for the task of process control results from its high resolution and high measuring speed, and from coaxial integration into the laser process, which allows for a spatially and temporally resolved representation of the weld bead topography during the process. To realize this, a spectral domain OCT (SD-OCT) system was developed and integrated into the beam path of the process laser. With the aid of suitable optics, circular scanning was realized, which allows for the 3D depth information to be displayed independently of the direction of movement of the processing head and the centrally running wire. OCT makes it possible to detect the process-typical topography deviations caused by process variations and thus paves the way for adaptive process control that could make additive laser processes more reproducible and precise in the future.

**Keywords:** additive manufacturing; LMD-w; optical coherence tomography; SD-OCT; inline; coaxial; process monitoring; beam shaping; multidirectional; circular scanning

## 1. Introduction

DED (Directed Energy Deposition) describes an additive manufacturing process in which thermal energy, usually supplied by a laser source, is used to melt the materials and deposit them layer by layer onto a substrate. Because of the good focusability of a laser source and the high degree of control over parameters like laser power, pulse duration, and wavelength, the laser as a processing tool offers the great advantage of precise energy deposition on the substrate. This makes the process suitable for manufacturing applications in a wide range of industries for example for tool making or tool repair [1]. Within the DED, mainly metal powders (e.g., in LPBF: laser powder bed fusion) or wires are used as feedstock material. The LMD-w process is considered particularly efficient and resource-saving due to the direct and complete deposition of the wire material used [2]. Unlike other processes, the LMD-w process is associated with less contamination and less health risks related to the use of metal powders, as well as a low-cost deposition [3]. To maintain precision during the entire process and to assess the quality of the welds, permanent process monitoring is required. Existing in-situ process monitoring is mainly limited to camera-based monitoring of the melt pool or indirect quality monitoring by means of temperature measurements. In

general, it can be stated that in-process workpiece monitoring is indispensable, since the occurrence of inhomogeneities, pores, or cracks, which can result from process instabilities, strongly influence the mechanical properties of the final product. A direct correlation of the qualitative characteristics to the process parameters, the basis of adaptive process control, is only possible with a spatially and temporally resolved assessment of these characteristics.

State-of-the-art wire processing heads use lateral feed of the process wire, which severely limits the flexibility of the system and does not allow for multidirectional applications. Asymmetric designs are also associated with more complex control systems and increased manufacturing times for the additive manufacturing of 3D components. For this reason, an LMD-w process with coaxial wire guidance was implemented. The difference between lateral wire guidance and a coaxial LMD-w process is illustrated in Figure 1. A measurement system based on low coherence interferometry was coaxially integrated into the process head, which allows for monitoring of the weld bead topography during the process.

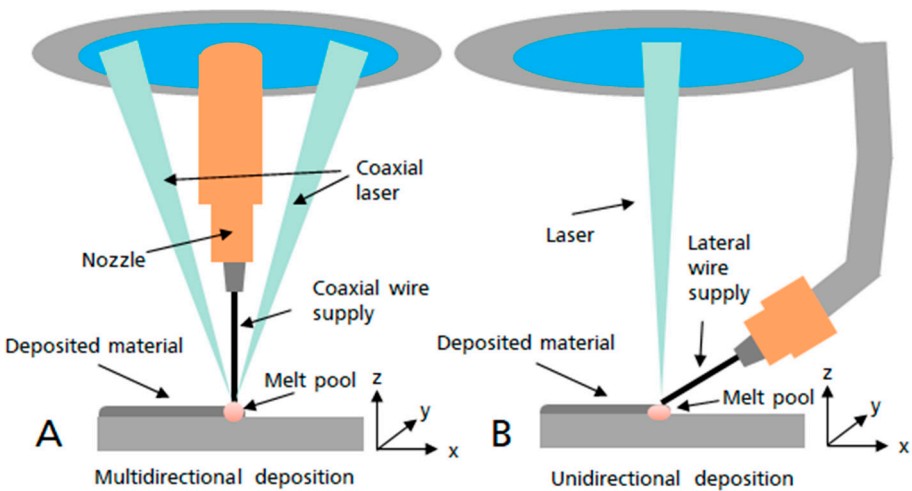

**Figure 1.** Schematic representation of a coaxial (**A**) and lateral (**B**) wire supplied system.

The measurement method is known as OCT; it originated in biomedicine and although it is still mainly used for retinal diagnostics, in recent years, OCT has become increasingly popular in industrial applications [4]. The basis of this method is an interferometer, with the Michelson interferometer-based set-up being the most commonly used. The beam of a low coherent light source (broadband spectrum) is split into two beam paths by a beam splitter; in the reference path, the light gets reflected by a mirror, while in the measuring path, the light gets reflected or backscattered from the surface of the sample or from deeper layers within a (semi-)transparent sample. Both reflected beams (from the reference path and measuring path) are recombined in the beam splitter. For a SD-OCT setup, as it is used in this work, the length of the reference mirror is fixed, and the depth information of the sample is encoded in the interference signal of the two recombined beams. The detection of the interference signal is realized by a spectrometer. The raw intensity signal detected by the spectrometer contains the information of the optical path length difference between sample path and reference path, encoded in the detected modulation signal. By performing a Fourier transformation on the detected intensity signal, the depth information of a single point measurement can be extracted. The Michelson interferometer setup can also be recognized in Figure 2, but instead of a bulk optic beam splitter, a fiber coupler is used, which serves the same purpose. The detailed functional principle of OCT, the most common signal processing steps and applications are described in detail in [5–7]. As a non-destructive testing method, OCT offers not only the advantage of non-contact and non-destructive inspection, but also a high spatial resolution and a fast measurement frequency. By using broadband light sources, it is possible to access in-depth information on (semi-)transparent components without the need for mechanical scanning in the z-direction.

Applications can be found for example in the characterization of optics [8], and in the quality inspection of polymers and ceramics [9,10]. Since the axial and lateral resolution of OCT are not coupled, samples with high aspect ratios can also be examined. Thus, the high precision of surface defect detection even at large working distances, the high measurement frequency, and the possible integration into laser processes make OCT a promising tool for in-process control of various laser material processing procedures, with the possibility of adaptive control based on spatially and temporally resolved acquisition of the workpiece to be processed. In this context, coaxial OCT has already been demonstrated for different laser surface structuring applications, e.g., as a non-destructive testing of micromachined surfaces with a so-called ultrahigh-resolution (UHR)-OCT [11] or for a dynamic in-line monitoring of structured electrical steel sheets using an additional pair of galvo mirrors [12]. OCT has also been tested for in-situ quality control in additive welding, e.g., by detecting the roughness of metal surfaces in powder-based deposition processes [13] or investigations into melt pool morphology dynamics in SLM (selective laser melting) [14]. Besides OCT, other methods of coaxial process monitoring have also been used for inline process control of laser processes. Integrated optical systems such as high-speed cameras, near infrared cameras, photodiodes, or thermal detectors such as pyrometers were used to assess welding quality from melt pool dynamics [15,16]. Although the data obtained by these methods can contribute to the understanding of the process, they do not provide direct information about the weld bead surface, which is essential for the quality of the final product. In-situ observation of the surface topography not only reveals qualitative characteristics, but also enables an extended understanding of topography deviations caused by varying process parameters. With the help of inline monitoring of the layered weld bead topography on the workpiece, process parameters can be adjusted based on the measurement data for stable process control.

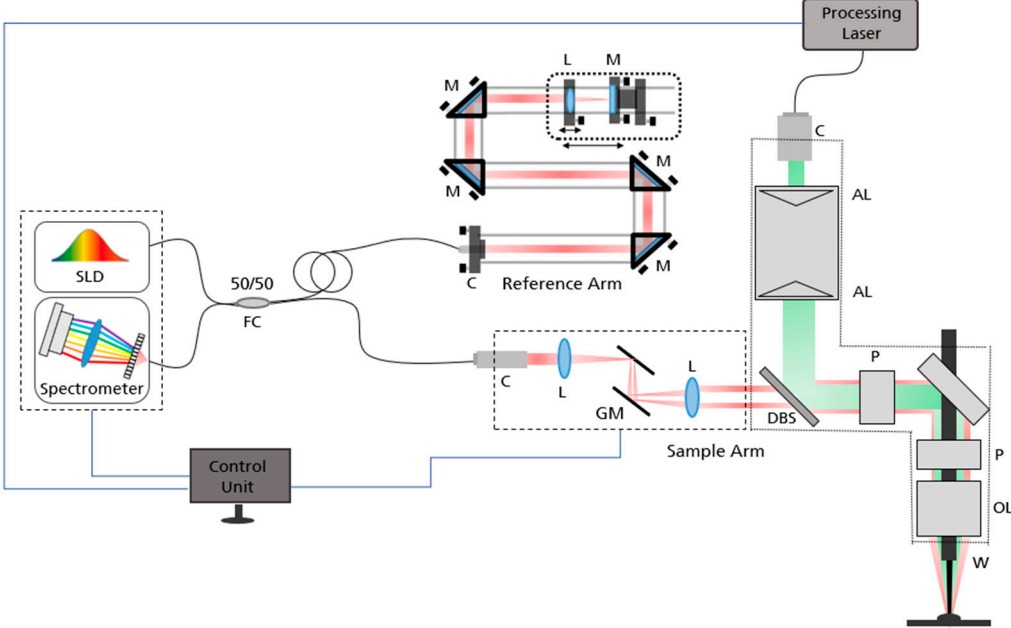

**Figure 2.** Schematic representation of the LMD-w system. It includes the coaxial processing head, the wire feeding system, and the integrated SD-OCT measuring system. (FC: Fiber coupler, DBS: Dichroic beam splitter, C: Collimator, OL: Objective Lens, M: Mirror, GM: Galvo mirrors, AL: Axicon Lens, P: Prism, W: Wire).

In this work, an advanced LMD-w system is presented and tested. The system contains an integrated, high-resolution SD-OCT for multidirectional in-situ monitoring of weld bead topography. An optical design developed for this application was used to overcome the challenges of coaxial wire guiding. It has been shown that the integrated OCT system can

detect shape deviations that would affect the quality and functionality of the workpiece. The use of OCT in the LMD-w process can eliminate the need for subsequent quality checks and significantly reduce process time and waste by adjusting process parameters during the process based on in-process measurement data.

## 2. Materials and Methods

The developed LMD-w-system consists of a laser process head, a coaxial wire-feeding system, the integrated OCT measuring system, and a control unit that controls all subsystems. Figure 2 shows the set-up, whereby it should be mentioned that the OCT components of the SLD (super luminescent diode), the spectrometer, and the reference arm are coupled to the process head by a single mode optical fiber. The length of the fiber was chosen so that the optical and electronical components were protected and placed in a distance from the welding process. The coaxial wire guidance of the system offers the advantage of multidirectional weld paths which can be realized, in contrast to lateral wire guidance, which adds flexibility to the process but creates some optical design challenges that were addressed in this work. To avoid diffraction effects and shielding by the wire, a beam shaping design using an axicon and optical prisms was chosen. Custom-built optics, such as the focusing lens, differ from ordinary optics in that their geometry allows for central wire guidance. To validate the optical setup, optical simulations were carried out, which verified a ring-shaped scan image on the sample. With the help of the simulations, the shape and arrangement of the optics could be determined. The shape of the conic optics significantly influences the shape of the laser beam focusing on the substrate in the processing area.

### 2.1. OCT Setup

The measurement system is based on SD-OCT technology. In SD-OCT, the frequency spectrum of the interference of back-reflected light from a sample arm and a reference arm with a stationary mirror is evaluated. The interferential spectrum contains the entire depth information of a scan in frequency-coded form, which means that axial scanning is not necessary and very high lateral scanning frequencies can be achieved. The SD-OCT developed for this work is based on a fiber-based Michelson interferometer setup consisting of three main components: Precitec's IDM (Precitec GmbH & Co.KG, Gaggenau, Germany), which combines a broadband light source and a customized spectrometer with a linear CCD camera, a scanning unit integrated in the LMD-w head with a dichroic beam splitter, and an external reference path that is shielded from any environmental influences and interference by the fiber-based coupling to the rest of the system.

A central wavelength of $\lambda_0$ = 1550 nm was chosen with a spectral bandwidth of 80 nm. In previous spectral emission tests, it was found that the process emissions have no influence on the interference signal of the measurement system. The light is guided through an optical fiber to a fiber coupler which splits the light in a ratio of 50/50 into the reference arm and the sample arm. The sample arm consists of a collimator, a telescopic beam expander, and a pair of galvo scanners. Inside the processing head, the measuring beam runs coaxially with the process laser after it passes the dichroic beam splitter; both the measuring beam and the process beam pass through optical prisms and the focusing lens, which focuses the beam onto the workpiece. The measuring range in the z-direction is 10 mm, the z-position of the workpiece can be determined with a repeatability of approx. 1 μm. The theoretical spot size in the focus plane was calculated to be 30 μm. A circular scanning pattern was necessary due to the presence of the coaxial nozzle and wire in the center of the objective lens. The circular scanning diameter of the measuring laser over the sample can be varied by adjusting the galvo parameters. In non-scanning mode, absolute height values can be read from the depth scan, the so-called A-Scan. With the use of the integrated galvo scanners, the measuring beam is scanned over the workpiece so that a 2D cross-sectional image of the sample is created. The A-Scan rate for the system used in this work was 70 kHz. A volume scan results from an additional scanning direction (e.g., in the direction of the process). For this application, a maximum scan field with a

scanning-diameter of 3 mm (r = 1.5 mm) was required to capture the complete width of the weld beads (shown in Figure 3). With a maximum galvo frequency of $\omega$ = 200 Hz and a measuring frequency of $f$ = 70 kHz ($t_{min}$ = 1.42857 $\times$ $10^{-5}$ s), the lateral spacing $\Delta Y_{max}$ can be calculated using

$$\left|\Delta Y_{max}\right| = r * \omega * t_{min} \tag{1}$$

to be ~4.3 µm. If the maximum feed rate $v$ of the process head is also considered, the spacing of the scanning points increases to 4.5 µm through additional path changes, using

$$\left|\Delta Y_{max}\right| = r * \omega * t_{min} + v\, t_{min}. \tag{2}$$

At the same time, the feed motion of the head causes the spot to be laterally washed out within the integration time of the camera of the spectrometer. This washing-out effect does not occur in conventional OCT because the measuring head or the object are not moved during acquisition. This effect plays a role when the lateral offset is greater than $\lambda/4$ [17]. With the integration time of the camera of 10 µs used here and a maximum feed rate of ~1000 mm/min, the lateral offset lies within this range. It was found that at feed rates of 1000 mm/min maximum, no contrast degradation of the signal is to be expected, so that the effect is negligible at these process speeds. If the process speed is to be increased by at least one order of magnitude, the integration time must be adjusted.

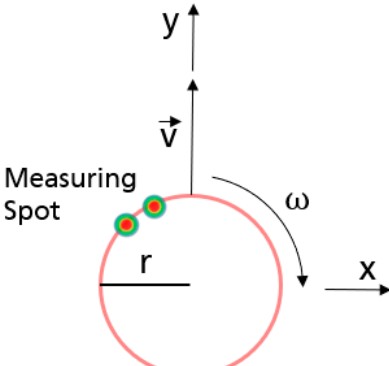

**Figure 3.** Scanning strategy of the OCT measuring laser showing two measuring spots on a circular path with defined radius *r*, which is moving with *v* in y-direction.

The theoretical resolution of the image information in the axial direction, which is reconstructed from the interference signal, is determined by the round-trip coherence length $l_c$ of the light source. The coherence length is defined by the central wavelength $\lambda_0$ and the spectral bandwidth $\Delta\lambda$ of the SLD [18]. Where the constant term $\gamma$ is 0.44 for a source for a Gaussian spectrum. However this term can differ depending on the windowing function used to sample the interferograms in the spectrometer [19].

$$\delta z = l_c = \gamma \frac{\lambda_0{}^2}{\Delta\lambda} \tag{3}$$

For the given case, this results in a theoretical axial resolution of 13.3 µm. The experimentally determined value of the actual axial resolution deviates from this value due to dispersion mismatch between the sample and reference arm and the use of different windowing functions. The FWHM of the depth scan was calculated to be 30.6 µm. Numerical dispersion compensation methods could be used to reduce peak broadening, which affects the measured FWHM of the depth, and thus the axial resolution.

### 2.2. LMD-w Process

The process head (Coaxprinter, Precitec GmbH &Go. KG, Gaggenau, Germany) is designed for high-performance processes. The coaxial wire-based laser metal deposition

system is shown in Figure 4. The laser beam is supplied by a diode laser source (Laserline GmbH, Mülheim-Kärlich, Germany). The diodes simultaneously emit four different wavelengths of 910, 940, 980, and 1030 nm. The available total laser power can reach a maximum of 5000 W, equally distributed over the four wavelengths. An optical fiber connects the laser beam to the processing head. The design of the head allows for coaxial wire guidance and omnidirectional process guidance. This means that the laser head does not have to be reorientated, as the orientation between wire and laser beam is independent from the feeding direction. In the experiments, IN718 wire from Quada V + F Laserschweißdraht GmbH, Hemer, Germany was deposited on flat S355 steel substrates.

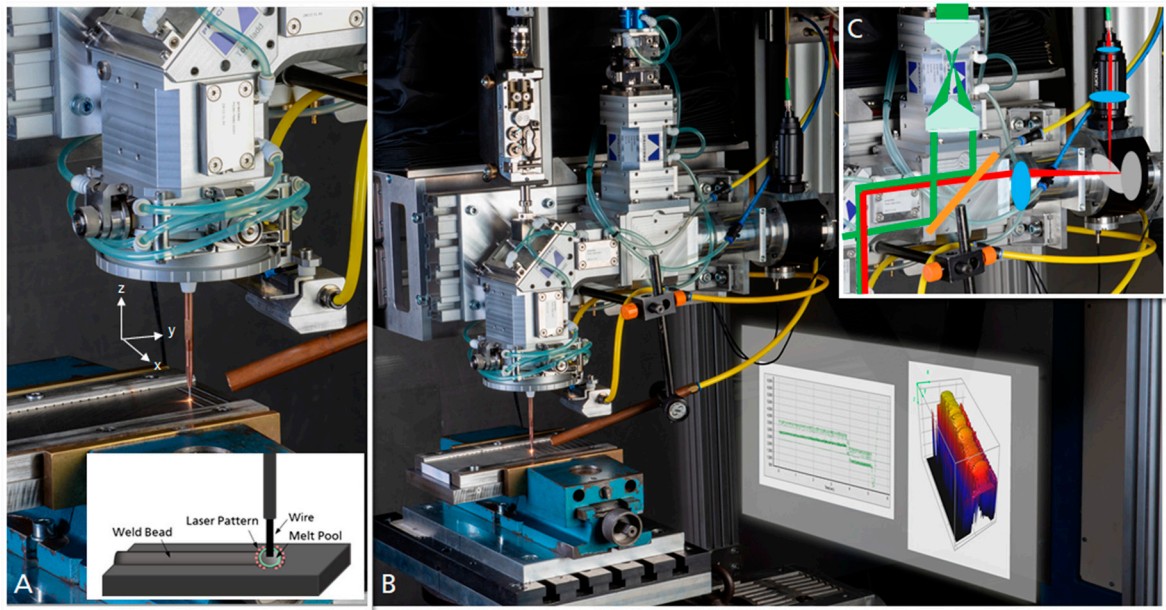

**Figure 4.** Coaxial wire-based laser metal deposition system (LMD-w) with an integrated optical coherence tomography (OCT) scanning head: (**A**) Coaxial wire supply head and schematic representation of the wire deposition with the circular laser pattern around the wire. (**B**) Overview of the entire system, including data visualization. (**C**) Superimposed schematic representation of the dichroic-mirror-based integration of the OCT scanning head and the double axicon lens module.

### 2.3. Beam Shaping Method

The greatest challenge in the integration of the measuring system into the LMD-w process lies in the aberration-free beam shaping of the measuring and process laser. In order to make the process directionally independent, a circular scanning around the coaxial wire was implemented. The minimum scanning-circle diameter of the measuring light is limited by the size of the wire or wire guide. The scanning field should also not be too large to be able to record enough data with adequate spacing at the maximum measurement frequency of 70 kHz.

For the required beam shaping, a combination of Axicon and prism is used in the setup, shown in Figure 5. A similar optical setup for beam shaping has already been shown in a previous work on the laser process of 3D laser brazing [20]. The original Gaussian distribution of the collimated laser beam is transformed into a ring shape by the Axicon. The additional material, in this case the wire, can be guided in the light-free center of the ring. To insert the coaxial wire guide into the ring, the ring is divided into two halves by a pair of refractive prisms. The arrangement of the prism provides a parallel offset of the incident beam, whereby an opening of the circle is realized. At one opening of the two resulting semicircles, the wire is inserted coaxially into the processing head. Finally, an aspherical lens focuses the process and measuring laser onto the processing zone. Here, the wire is melted with the help of the process laser, resulting in a deposition on a workpiece. This functional principle reveals the importance of the optimal distance

between the focusing position of the process laser and the end of the wire. The optics used, such as optical prisms and Axicon, allow annular beam guidance, and thus flexible control of the LMD-w process head, which is independent of the measuring system.

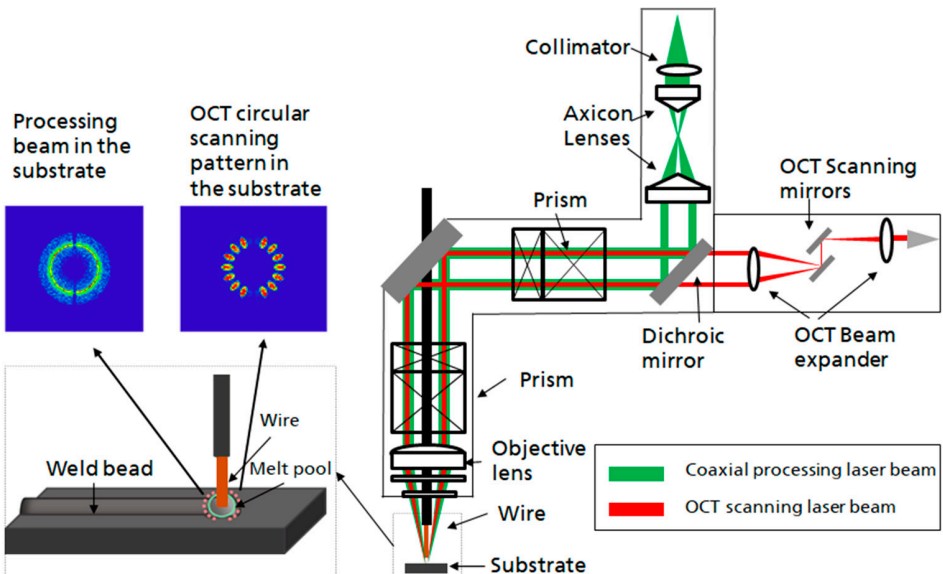

**Figure 5.** The optical set-up shows the optical paths of the processing and scanning beam from their corresponding collimators to being focused on the substrate. The blue image planes show the spot diagrams of the focused processing beam and the circular scan pattern of the focused measurement beam in the melt pool.

*2.4. Signal Processing*

The superposition of the light reflected by the sample and the light reflected from the reference arm results in an interferogram, which is detected with the spectrometer. To be able to read depth information from the raw signal, a fast Fourier transformation (FFT) is performed on the measured spectrum by means of interpolation after conversion into wavenumber units. The height information in the generated A-Scan is not given by the absolute intensity information of the backscattered light, meaning that a stable signal can still be detected even in the case of larger fluctuations in the process that can influence the proportion of backscattered light. Figure 6a shows the interference spectrum, which is a combination of the Gaussian shaped intensity spectrum of the light source and the overlaying modulation signal. Figure 6b shows the corresponding A-Scan. In the case of highly reflective metal surfaces, the height of the peak can be well extracted from the background noise and provides the absolute distance to the sample surface. With the help of the scanning by the galvo mirrors used and the relative movement of the process head to the weld bead, a volume scan, the geometric profile, is created. In order to eliminate any remaining signal noise, a dark reference is carried out before the measurement process. For this purpose, the sample arm is blocked, and the resulting signal is subtracted from the actual signal during the measurement process. The A-Scans recorded during the process are additionally filtered with a quality threshold value in order to reduce the further noise.

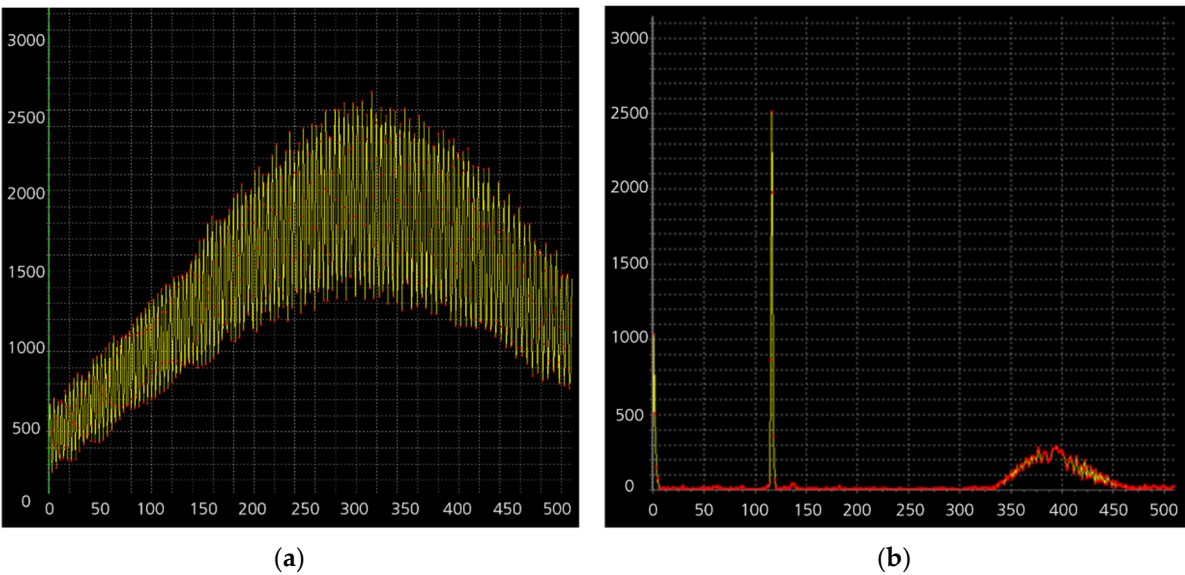

(**a**)     (**b**)

**Figure 6.** OCT measuring signal (**a**) interference spectrum showing the Gaussian shape of the SLD. (**b**) Depth information resulting from the interference via FFT (A-Scan) with a FWHM of ~30.6 μm.

## 3. Results

### 3.1. LMD-w Process Parameters

In a parameter study, going along with visual inspection and optical geometry measurements of the beads, suitable process parameters were identified. In a first step, the laser power was varied while the wire feeding rate and the travel speed were kept constant, at 1080 and 900 mm/min, respectively (ratio vW/vM = 1.2). It was shown that laser powers below 850 W were not sufficient to provide enough energy for complete wire melting. Wire scratching through the melt pool and bonding defects occurred. However, above 1650 W, the high energy input led to fume formation and strong oxidation of the bead surface. In a second step, the laser power was kept constant at 1400 W and the travel speed at 900 mm/min, while the vW/vM ratio was increased from 1.1 to 1.65 (corresponding to wire feeding rates from 990 mm/min to 1485 mm/min). At a ratio lower than 1.1, the fed wire quantity was insufficient to enable a steady state and homogeneous deposition. The beads show discontinuities. Above 1.65, scratching occurred, as the laser power was too low to melt the comparatively high wire volume. High speed imaging of the process allowed us to analyze the melt pool behavior and the interaction between wire and melt pool.

It was observed that high energy input (either caused by high laser power or low travel speed) leads to a large melt pool. In this case, the randomly occurring melt pool oscillations are more pronounced and cause the wire–melt pool interaction, thereby making the process more unstable.

After the depositions, the bead cross-sectional geometry was determined by measuring the workpiece with an offline OCT-system. The results are shown in Figure 7. It is concluded that among the varied parameters, the laser power has the strongest influence on the bead geometry. The bead width varies between 1.5 and 2 mm in the considered parameter range. The height changes from 1.1 to 0.75 mm, respectively. For the experiments with the coaxially integrated OCT-system, the following parameters were selected which delivered reproducible and satisfactory results: PL = 1400 W, vM = 900 W, vW = 1170 W.

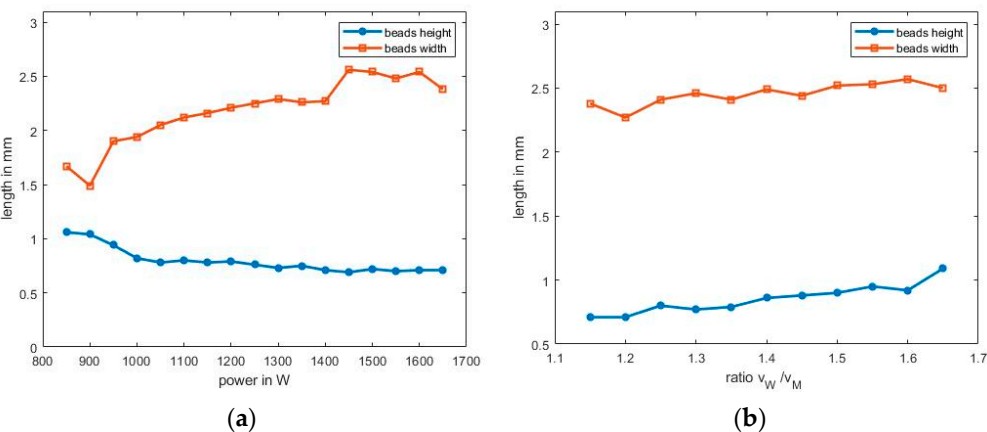

(**a**)　　　　　　　　　　　　　　(**b**)

**Figure 7.** Analysis of the weld bead geometry (**a**) in dependence on the laser power and (**b**) in dependence on the ratio vW/vM.

### 3.2. Offline Measurements

After system integration, the functionality of the coaxially integrated OCT-system was verified in offline mode. This means that the process laser was not active, and thus effects of the laser process on the measurements could be ruled out. The testing of the functionality of the coaxially integrated OCT-system included scanning the surface topography of the substrate to determine the distance to the workpiece and check the weld bead quality. In the simplest case, in A-Scan mode, the coaxially integrated OCT-system was used to adjust the distance of the workpiece to the exact focus plane of the system. Depth calibration of the measuring system was carried out by using a stepped calibration target (Figure 8); through calibration, the absolute distance of the substrate within the measuring depth, in the positive and negative z-directions outside the focus, can be determined with particular precision. Previous work has already shown that the optimum focus distance of the process laser plays a critical role in process stability. The distance influences the size of the ring-shaped laser beam in the processing plane and thus also the amount of energy transferred to the wire. The effects on the weld are referred to as "stubbing" when the energy transfer is insufficient, whereas "dripping" occurs when the distance between the melting wire and the substrate is too large [21,22]. The focal distance of the measuring laser corresponds to the focal distance of the process laser, since both light sources share the same focusing optics. Therefore, these effects can be avoided if the distance of the workpiece is monitored using the coaxially integrated OCT-system.

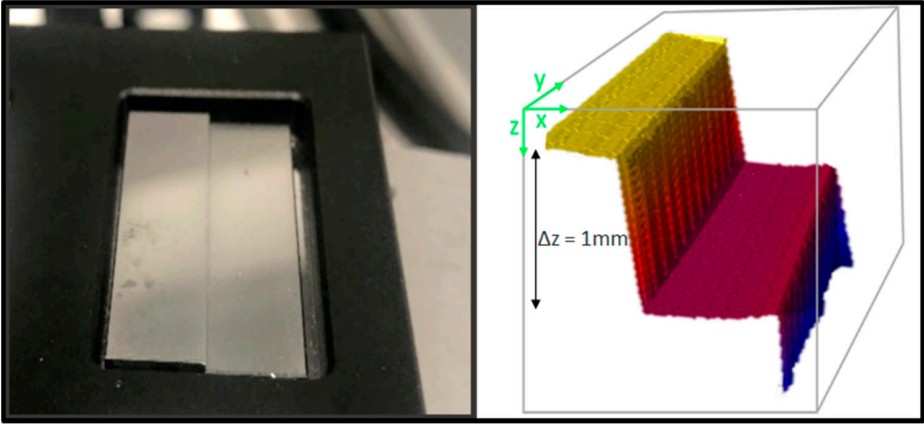

**Figure 8.** Picture and OCT volume scan of a step calibration target.

### 3.3. Inline Measurements

The functionality of the coaxially integrated OCT-system was also tested during the process. No significant differences could be found in the OCT images of the welds taken in offline mode when compared to the images taken in inline mode. This means that the process laser has no influence on the inline measuring system or the quality of the measurement images. By varying the process parameters (laser power, feed rate, and wire feed speed), the influence of these parameters on the measured OCT-signal was evaluated. Two additional measurement systems were used to validate the data acquired with the coaxially integrated OCT-system: A high-speed camera was integrated into the measurement setup to observe the melting process (Figure 9b) and a commercial offline tabletop OCT system (Thorlabs, Newton, NJ, USA) was used to qualitatively verify the measured weld bead dimensions after the process (Figure 9a).

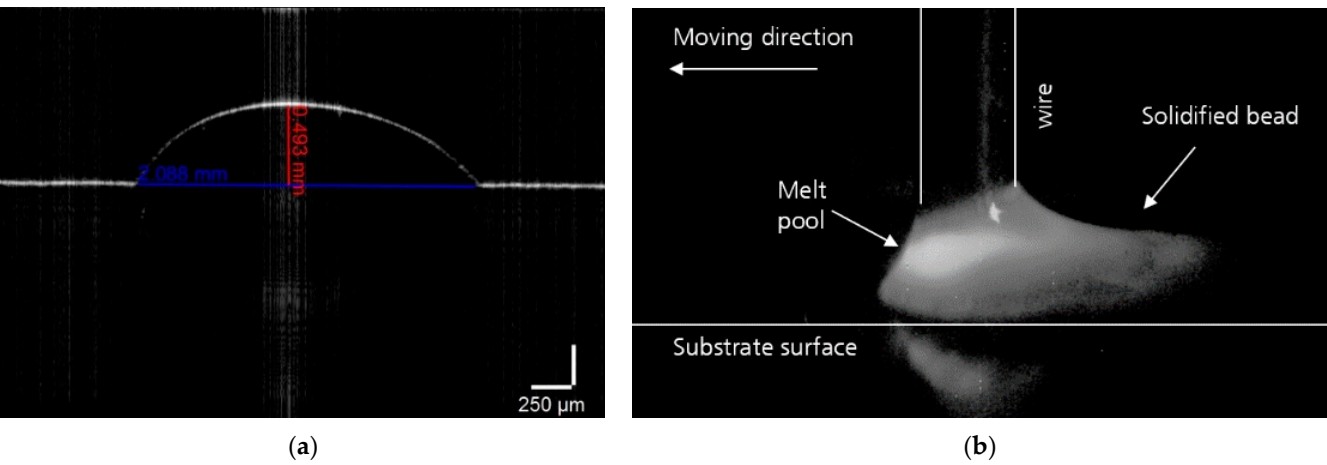

(**a**)　　　　　　　　　　　　　　　　　　　　　　　　　　(**b**)

**Figure 9.** (**a**) OCT B-Scan of a weld bead acquired with a commercial tabletop system. (**b**) High speed camera image of the weld bead acquired in the process.

Once the wire height and focus distance were set, welds were generated on a substrate surface as test samples. During the laser process, OCT data were acquired with the coaxially integrated OCT-system and post-processed both as 2D data (B-Scans) and 3D data (C-Scans). The data then were compared with the data acquired after the process with the offline OCT system and through simple visual observation of macroscopic shape deviations. The OCT images, which were acquired with the coaxially integrated OCT-system, show shape deviations that typically occur in the LMD-w process. Figure 10A,B show clear deviations from an optimum weld bead in the form of ripples at various positions. At the starting point of the process, a uniform and smooth weld bead was generated (Figure 10B left). In the course of the process, the distance (observed in the OCT B-Scan, Figure 10C) from the workpiece surface to the end of the process head changed, resulting in the afore-mentioned shape deviations. The OCT images (Figure 10) show these deviations, which are clearly visible and quantifiable in the volume Scan acquired by the coaxially integrated OCT-system. Within the inline measurements, numerous welds with different wire diameters were tested up to a maximum width of the resulting welds of 3 mm.

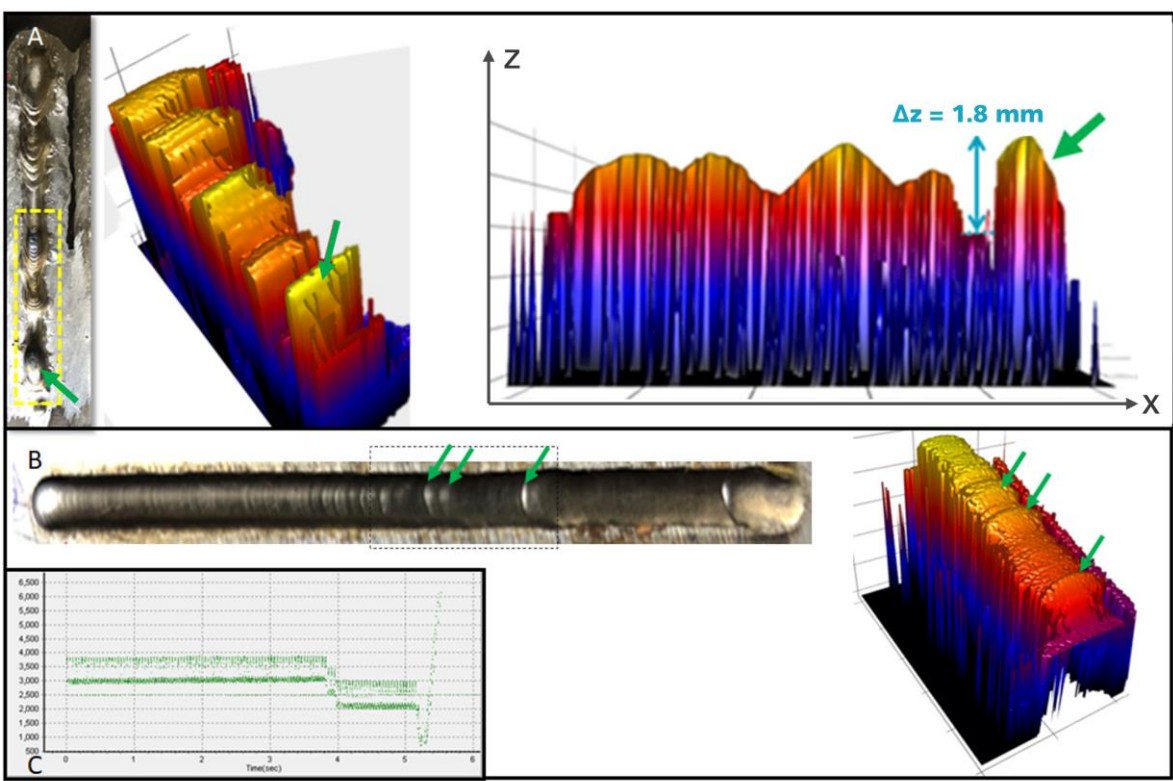

**Figure 10.** (**A**) Picture and OCT Volume and B-scans of a weld bead showing high uniformities with up to 1.8 mm differences in z-direction (**B**) Picture and volume scan of a weld bead showing form deviations. (**C**) OCT-B-Scan showing changes in z-position (distance between processing head and weld bead).

## 4. Conclusions and Discussion

The coaxial LMD-w process is considered a promising method for future additive manufacturing processes. Depending on the wire size, the LMD-w process can be used in a variety of applications, from the production of 3D micro components and complex 3D structures to the use of LMD-w for the repair of existing tools or components of any dimension. Especially for the use in safety-relevant environments, such as aircraft construction, the tolerance limits for shape deviations are tight. In order to meet those high-quality standards while forming a 3D structure from a single weld bead, inline topography monitoring is required to react to deviations of the additively manufactured structure. The quality of the individual layers is particularly decisive for the strength and has a considerable influence on the function of the manufactured component. To ensure sufficient quality control and to continuously monitor the process, OCT was successfully coaxially integrated into an LMD-w process for the first time, as part of this work. It could be shown that the weld bead topography can be captured within the running LMD-w process with high measurement frequency and high spatial resolution. In the inline OCT measurements of individual weld beads, the occurrence of waviness, shape deviations, or other defects could be resolved.

One core aspect of the developed system is its flexibility. The flexibility of the system is achieved by the combination of two key concepts: the coaxial LMD-w process head itself with a centrally guided wire, which allows for movement of the process head in any direction and adaptive circular scanning of the process lasers, and, in particular, the measurement laser, which allows for in-process topography control. In combination, these two systems allow for what is called in the context of this work multidirectional process control of the LMD-w process. In A-Scan mode, the distance of the optics to the substrate surface can be quantified and set with an accuracy of ~1 μm before the process and monitored over

the duration of the process. In 3D mode, the weld bead topography including defects can be captured. In offline mode (without active process laser), the substrate surface can be scanned and qualitatively evaluated. This is particularly helpful when the LMD-w process is to be used to repair existing workpieces. The workpiece is scanned enface in 3D mode and its defects are detected with the OCT. The repair process planned on the basis of the measurement data can then be monitored in inline measuring mode.

The described attributes distinguish the method presented in this work from other coaxial monitoring systems of laser welding processes. It has been described that coaxial camera monitoring can be used to assess melt pool expansion within the process [23]. However, a topography of the final weld bead cannot be extracted using this method, which means that defects such as pores or similar phenomena can remain hidden from the measurements, which makes the coaxial integration of a camera for melt pool control another indirect process monitoring solution. Laser triangulation has been used coaxially to assess the height in a powder-based LMD process [24]. The big advantage of the approach of this work to integrate OCT in the process is the ability of OCT to measure geometries with high aspect ratios and challenging surfaces (high or low reflectivity, high surface roughness), in cases where laser triangulation would reach its limits. Additionally, OCT can reach much higher spatial resolutions in comparison to laser triangulation, that makes an LMD-w process monitored with OCT also suitable for very demanding production scenarios of small scale products.

Based on the shown measuring concept, adaptive process control can be developed by coaxial integration of the OCT measurement system. In the future, topography deviations or defects can be automatically detected and classified with trained networks and the process can be controlled based on in-process data. OCT also offers the advantage of being a non-destructive measurement method capable of generating a large amount of data in a very short time, which could be the gateway for applications in artificial intelligence for additive manufacturing.

**Author Contributions:** Conceptualization, C.S., M.K.-H., A.V.I.; methodology, C.S. and M.K.-H.; software, M.K.-H.; validation, M.G. and C.S.; formal analysis, C.S.; investigation, C.S. and M.K.-H.; resources, C.S. and M.K.-H.; data curation, C.S.; writing—original draft preparation, C.S..; writing—review and editing, C.S., A.V.I., M.G., M.K.-H. and R.H.S.; visualization, C.S. and A.V.I.; supervision, R.H.S.; project administration, C.S.; funding acquisition, R.H.S. All authors have read and agreed to the published version of the manuscript.

**Funding:** This research was funded by the German Federal Ministry of Education and Research within the funding program Photonics Research Germany in the project TopCladd (Grant number of 13N14265).

**Institutional Review Board Statement:** Not applicable.

**Informed Consent Statement:** Not applicable.

**Acknowledgments:** The authors acknowledge all project partners of the research project TopCladd for their support. This work was supported by the Belgian partners Deltatec and LaserCo Dt and the German project partners Precitec, Quada V + F Laserschweißnaht GmbH and Dinse GmbH.

**Conflicts of Interest:** The authors declare no conflict of interest.

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
