# Peer review of "Inline Optical Coherence Tomography for Multidirectional Process Monitoring in a Coaxial LMD-w Process"

_applsci, doi:10.3390/app12052701_

Round 1

Reviewer 1 Report

The authors present an application for an FD-OCT attached in the coaxial form to an LMD-w system. The application is interesting, but there are some concerns,

  1. Line 2, ¨Omnidirectional…¨ not sure. Maybe circular will fit better with the actual scanning process.
  2. Line 15, LMD-w is mentioned but not defined. Only at line 30 is it defined.
  3. Line 69, UHR-OCT is not defined.
  4. Line 84, ¨novel LMD-w…¨ not fancy with words like novel, in applications. Both methods are known, ring illumination also.
  5. The optical setup is poorly described. As this is an application, more experimental information is desired. The OCT scanning and the coaxial matching are relevant in this work, and they are only briefly mentioned. Please enhance this section.
  6. Equation 3 is a general approximation in the air for depth resolution. The use of filters and processing windows affects the constant term. This term could be referenced in: “Double-shot depth-resolved displacement field measurement using phase-contrast spectral optical coherence tomography,” Optics Express, 14(21) 2006, to avoid this ambiguity.
  7. Figure 4 is hard to understand with those gray cylinders and blocks. The coaxial configuration is one of the innovations of this work. It should be explained in detail.
  8. Figure 3 axes are impossible to see. Information about the sensor is also omitted. The latter affects the available depth range (comment 6).
  9. Section 3.3. Slightly confused, during the melting, are both OCT systems used? The first paragraph indicates that a commercial Oct was used only after the process, but it later suggests that both systems were used during the process. If a commercial OCT could image during the melting… why use a complex coaxial Oct? Please clarify this.

Author Response

Dear Reviewer,

Thank you very much for the helpful and important comments on our manusprit!

we have revised and optimised the manuscript again with the help of your comments, below you will find the answers to your points.

Many thanks and best wishes

Charlotte Stehmar

Point 1: Line 2, ¨Omnidirectional…¨ not sure. Maybe circular will fit better with the actual scanning process.

Response 1: it is changed to “multidirectional”, which is derived from circular scanning and relates to both the measurement system and the process as a whole.

Point 2: Line 15, LMD-w is mentioned but not defined. Only at line 30 is it defined.

Response 2: LMD-w is now also defined in line 15, and we have also added more information about LMD-w in the first part of the introduction.

Point 3: Line 69, UHR-OCT is not defined.

Response 3: Done, now line 75.

Point 4: Line 84, ¨novel LMD-w…¨ not fancy with words like novel, in applications. Both methods are known, ring illumination also.

Response 4: Done, now line 96.

Point 5: the optical setup is poorly described. As this is an application, more experimental information is desired. The OCT scanning and the coaxial matching are relevant in this work, and they are only briefly mentioned. Please enhance this section.

Response 5: We added more information to the setup and also renewed the figure with the optical setup so that the circular scanning will become more clear. 

Point 6: Equation 3 is a general approximation in the air for depth resolution. The use of filters and processing windows affects the constant term. This term could be referenced in: “Double-shot depth-resolved displacement field measurement using phase-contrast spectral optical coherence tomography,” Optics Express, 14(21) 2006, to avoid this ambiguity.

Response 6: We changed the formula and also addes the reference you sent.

Point 7: Figure 4 is hard to understand with those gray cylinders and blocks. The coaxial configuration is one of the innovations of this work. It should be explained in detail.

Response 7:  We added a new figure (figure 5) so that the coaxial configuration come out more clear

Point 8: Figure 3 axes are impossible to see. Information about the sensor is also omitted. The latter affects the available depth range (comment 6).

Response 8:  We also renewd figure 3 (it is now figure 4) and figure 6 with new axes and added some inforamtion on the spectometer line 137.

Point 9: Section 3.3. Slightly confused, during the melting, are both OCT systems used? The first paragraph indicates that a commercial Oct was used only after the process, but it later suggests that both systems were used during the process. If a commercial OCT could image during the melting… why use a complex coaxial Oct? Please clarify this.

Response 9: We added some more information and also changed the wording. The commercial OCT was only used for some hight measurements after the process to characterize the process itself (see figure 7). With the commercial OCT it is not possible to measure heights during the process.

Reviewer 2 Report

The MS reports an innovative study in which a new optical monitoring system applied to the wire-based laser metal deposition LMD-w process is presented and tested.  The results showed that the developed monitoring system, specifically high-resolution SD-OCT (spectral domain optical coherence tomography), is able to detect shape deviations in process, which would negatively affect the final quality of the component. In addition, a calibration of the system has been performed to ensure the validity and repeatability of the results shown by varying the different process parameters (laser power, feed rate and wire feed speed). Therefore, based on the experimental data presented, and the well justified argument the MS is worthy of publication in applies after a minor revision.

Only a few suggestions can be made to improve and make the paper complete:

1. p. 4 line 155-159: You could better explain this negligible " washing-out effect ". Actually this effect could be a significant criticality for the implemented monitoring system. How did you manage to show that within 1000 mm/min the effect is negligible? What suggestions do you have for higher scan speeds?

2. p. 9 line 293: Could you better describe the image analyses that have been carried out, describing the algorithms used to perform the measurements described in Fig. 9? 

The final section of the paper is called “Discussion and outlook“. I think this section is more appropriate as “Conclusions”. Instead, I suggest inserting a “Discussion” section after “Results” in order to better discuss obtained results. Moreover, in literature, there are several works on coaxial optical monitoring systems that investigate outputs similar to this work. I think authors should extend and compare their results with those in the literature and bring out the advantages of this methodology.

Below I suggest some papers that authors can add  to the discussion:

[1]       Errico, V., Fusco, A., Campanelli, S.L. Effect of DED coating and DED + Laser scanning on surface performance of L-PBF stainless steel partsSurface and Coatings Technology, 2022, 429, 127965. 

 [2]      Errico, V., Campanelli, S.L., Angelastro, A., Dassisti M., Mazzarisi, M., Bonserio, C., Coaxial monitoring of aisi 316l thin walls fabricated by direct metal laser deposition, Materials, 2021, 14(3),673, pp. 1-17.

[3] Donadello S., Motta M., Demir AG,  Previtali B., Monitoring of laser metal deposition height by means of coaxial laser triangulation, Optics and Lasers in Engineering Volume 112, January 2019, Pages 136-144.

Author Response

Dear Reviewer,

Thank you very much for the helpful and important comments on our manusprit!

we have revised and optimised the manuscript again with the help of your comments, below you will find the answers to your points.

Many thanks and best wishes

Charlotte Stehmar

Point 1: p. 4 line 155-159: You could better explain this negligible " washing-out effect ". Actually this effect could be a significant criticality for the implemented monitoring system. How did you manage to show that within 1000 mm/min the effect is negligible? What suggestions do you have for higher scan speeds?

Response 1: The part about the lateral washing-out ( now line 179-188) was adapted with much more information and a calculation. Also a suggestion for even higher process times was given. At very high feed rates the lateral resolution would deteriorate due to the smearing of the spot if the integration time of the camera remained the same. Studies on this are still pending

Point 2: p. 9 line 293: Could you better describe the image analyses that have been carried out, describing the algorithms used to perform the measurements described in Fig. 9? 

Response 2: We added more information about the OCT setup itself and described the image analysis (fast fourier transform) also we mentioned how to process the A-Scans into volume scans by scanning over the sample.

Point 3: The final section of the paper is called “Discussion and outlook“. I think this section is more appropriate as “Conclusions”. Instead, I suggest inserting a “Discussion” section after “Results” in order to better discuss obtained results. Moreover, in literature, there are several works on coaxial optical monitoring systems that investigate outputs similar to this work. I think authors should extend and compare their results with those in the literature and bring out the advantages of this methodology..

Response 3: We adjusted the the last chapter with your comments and also addes the refereces you sent. It is now called “Conclusion and discussion” so the first part is a summery of the results, wich was missing in the first draft. After the conclusitn there is a discussion part with references to other coaxial system (triangulation and camera-based).

Reviewer 3 Report

  1. This paper describes an inline monitoring system for LMD-w. The LMD and OCT are mature technologies, but the authors combine the two technologies to explore the new application field. The organization of this manuscript is fine and brief. However, some suggestions below are provided for the authors' reference.
  2. Because this research is application-based, the readers might come from different research fields. Could the authors provide more references about the LMD and OCT for a better understanding of them?
  3. Could authors make a quantitative comparison between offline and inline measurements? In this way, the readers can better understand the capability of the proposed system.
  4. The qualities of all of the figures are too low to discern. Please revise them.

Author Response

Dear Reviewer,

Thank you very much for the helpful and important comments on our manusprit!

we have revised and optimised the manuscript again with the help of your comments, below you will find the answers to your points.

Many thanks and best wishes

Charlotte Stehmar

Point 1: Because this research is application-based, the readers might come from different research fields. Could the authors provide more references about the LMD and OCT for a better understanding of them?

Response 1: we have adopted this part in the introduction. First the LMDw is described again and in the second part the OCT (with references).

Point 2: Could authors make a quantitative comparison between offline and inline measurements? In this way, the readers can better understand the capability of the proposed system.

Response 2: We have summarised this part again in the "conclusion" part. It is now clear what can be achieved in inline mode and what can be achieved in offline mode. And also why the inline mode is so important .

Point 3: The qualities of all of the figures are too low to discern. Please revise them

Response 3: Done, we have updated most of the figures with more detailed information and also added some additional figures.